# ALIGNMARK: CONTENT-ALIGNED AUDIO WATERMARKING FOR ROBUSTNESS AGAINST NEURAL TRANSFORMATIONS

## ABSTRACT

Audio watermarking, which embeds identity information into audio for authenticity verification, is an effective approach to protecting the intellectual property of audio content creators. A key unresolved challenge in audio watermarking is the limited robustness of existing methods under real-world neural transformations, such as denoising, codec, and vocoder reconstruction, which can render watermarks removable or undetectable. To better understand this challenge, we introduce the content alignment degree (CAD) metric, which quantifies the extent to which watermarks are integrated into audio, and observe a positive correlation between CAD and watermark robustness. Guided by CAD, we propose AlignMark, a content-aligned audio watermarking method that leverages spectral masking in the embedder, temporal masking in the decoder, and multiple perceptual losses to explicitly align watermark embedding with audio content and improve robustness against diverse attacks while preserving perceptual quality. Furthermore, a feature pyramid-based decoder extracts watermarks across multiple scales, enhancing reliability under pitch shifts and spectral distortions. Extensive experiments on multiple datasets and 21 attack scenarios demonstrate that AlignMark achieves state-of-the-art performance, with an average bit-wise accuracy of 0.98 and false attribution rate of 0.05, while maintaining imperceptible impact on audio quality. See our code and demos at: https://anonymouswatermark.github.io/alignmark/.

## 1 INTRODUCTION

Audio watermarking, which embeds identity information into audio for authenticity verification, is an effective approach for protecting the intellectual property (IP) of audio content creators. Robustness is a key requirement, as it determines whether embedded information can survive real-world attacks and distortions; without it, watermarks become removable or undetectable, limiting practical utility. Traditional methods, such as spread spectrum (Bender et al., 1996), echo hiding (Gruhl et al., 1996), and quantization index modulation (Chen & Wornell, 2001), have been studied for decades, but their limited robustness against complex attacks restricts practical adoption. More recently, deep neural network (DNN)-based audio watermarking (Chen et al., 2023; San Roman et al., 2024; Liu et al., 2024a; Li et al., 2025) has demonstrated substantial improvements in robustness.

Despite progress in DNN-based audio watermarking, existing research remains nascent, addressing limited attack scenarios and focusing mainly on robustness against traditional audio distortions like resampling, filtering, and compression. Studies (O'Reilly et al., 2025; Wen et al., 2025) highlight the lack of robustness in current approaches against complex neural transformations, including denoisers (Zhao et al., 2025), codecs (Défossez et al., 2022; Ju et al., 2024; Zhang et al., 2024), and vocoders (Kong et al., 2020; Siuzdak, 2024); However, the underlying causes of robustness degradation remain unexplored, and no existing solutions have effectively addressed this issue.

An intuitive factor of this issue, as identified by O'Reilly et al. (2025), is that previous methods embed watermarks as background artifacts, decoupled from audio content, which makes them vulnerable to removal by denoising or codec reconstruction. This finding motivates us to propose a further insight: *Robustness correlates with the degree to which watermark embedding is aligned with the audio content, or the extent to which the watermark is contained within the audio content.*

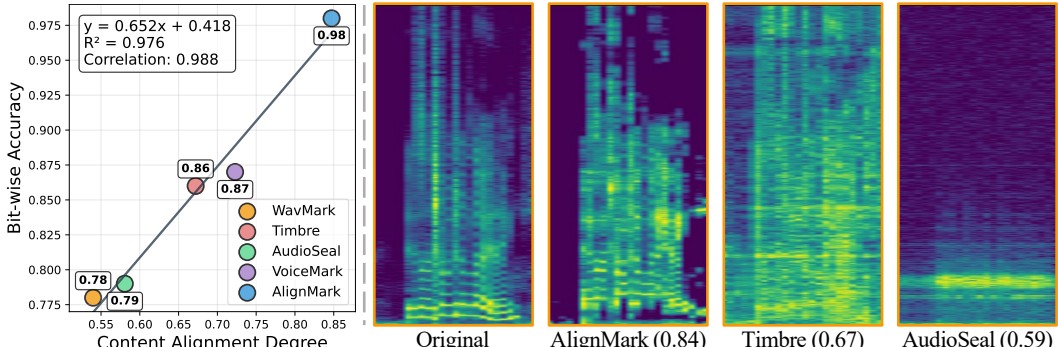

Figure 1: **Left**: content alignment degree vs bit-wise accuracy. **Right**: examples of original audio and watermark spectrograms for methods (content alignment degree values in parentheses)

Here, audio content can be broadly interpreted as the energy-dense regions in the spectrogram. For instance, in the case of speech, an ideal watermark should integrate into the harmonics and formants, which regions are typically preserved under attacks, thereby providing inherent robustness. To validate this insight, we formulate a metric called content alignment degree (CAD), which quantifies the extent to which the watermark is contained within the audio content. Specifically, we treat the spectrogram as an image and measure the watermark coverage within audio content (energy-dense regions) across frames and frequency bands. Figure 1 (left) illustrates the relationship between CAD and bit-wise accuracy (ACC) across multiple watermarking methods. The results reveal a strong positive correlation between CAD and ACC of various methods, providing empirical evidence that the degree of content alignment is crucial to the robustness of the watermark.

As shown in Figure 1 (right), we visualize the spectrograms of watermarks generated by multiple methods. By comparing the spectrograms of watermarks with the original audio, previous methods exhibit limited alignment with the audio content. From a model perspective, most existing methods do not explicitly optimize for the degree of content alignment (Chen et al., 2023; San Roman et al., 2024; Liu et al., 2024a; 2025). Instead, they typically adopt general architectures that embed watermarks into audio globally, without distinguishing between silent segments and voiced content. While Li et al. (2025) considers embedding watermarks into voiced frames, it lacks constraints in the frequency domain, resulting in partial watermarks in non-content frequency bands and degraded audio quality. These limitations compromise their robustness against complex attacks.

In this paper, we propose AlignMark, a novel content-aligned audio watermarking method that explicitly aligns watermark embedding with audio content to provide inherent robustness. As discussed earlier, CAD measures the watermark alignment across frames and frequency bands, which motivates the design of AlignMark to focus on alignment in both temporal and spectral dimensions. Therefore, AlignMark incorporates spectral masking for the watermark embedder, temporal masking for the watermark decoder, and leverages multiple perceptual losses to guide the model toward content-aligned watermarking. Specifically, temporal masking leverages voice activity detection (VAD)-based loss to directly constrain watermark decoding to voiced frames, while perceptual losses enforce consistency between the watermarked audio and the original audio in both the time-frequency domain and acoustic features. The gradients derived from these losses are then back-propagated to the embedder, where the spectral masking explicitly aligns the watermark with the audio content on the spectrogram. Additionally, inspired by Wen et al. (2025), which observes that previous watermark decoding heavily relies on fixed frequency bands, making it susceptible to pitch shifting, we introduce a feature pyramid in watermark decoding to extract watermarks across multiple scales and improve robustness. Our main contributions are summarized as follows:

- We propose the content alignment degree (CAD) metric to quantify the alignment between watermarks and audio content in the spectrogram. CAD reveals a positive correlation between watermark robustness and alignment with audio content, providing a guiding perspective for developing robust audio watermarking against neural transformations.

- We propose AlignMark, a novel content-aligned audio watermarking method that explicitly aligns watermark embedding with audio content in both the temporal and spectral dimen-

sions. Combined with multiple perceptual losses and a feature pyramid-based watermark decoder, AlignMark achieves inherent robustness against complex attacks.

- Extensive experiments on multiple datasets across 21 attack scenarios, including denoising, codec, and vocoder reconstruction, demonstrate the robustness of AlignMark. It achieves an average bit-wise accuracy of 0.98, surpassing state-of-the-art methods at 0.87, while maintaining imperceptible audio quality impact for human listeners.

## 2 RELATED WORK

We classify related work into two types based on practical scenarios: general and generative. General audio watermarking, the focus of this paper, integrates watermarks into existing audio, whether artificially created or naturally recorded. Generative audio watermarking embeds watermarks during audio generation, ensuring all generated content inherently contains identifiable marks.

**General Audio Watermarking**. Traditional methods, such as spread spectrum (Bender et al., 1996), echo hiding (Gruhl et al., 1996), and quantization index modulation (Chen & Wornell, 2001), have been studied for decades. Recently, DNN-based audio watermarking has emerged with diverse approaches: spectrogram-based methods embed watermarks in spectral representations (Chen et al., 2023; Liu et al., 2024a); waveform-based methods generate watermark waveforms that are directly added to the original audio (Li & Lin, 2024; San Roman et al., 2024); synthesis-based methods leverage pre-trained codecs to directly generate watermarked audio (Li et al., 2025; Ji et al., 2025). While these DNN methods outperform traditional approaches, they employ general architectures without audio content alignment, and still face challenges against complex neural transformations. To evaluate our proposed content-aligned watermarking, we comprehensively compare it against the aforementioned methods using publicly available implementations.

**Generative Audio Watermarking**. With the advancement of generative AI, some studies have begun exploring watermarking methods embedded within generative models, enabling audio content to carry watermarks inherently without requiring additional processing. Some approaches design watermark-enabled codec models, allowing autoregressive generative models trained with the codec's tokens to produce audio with built-in watermarks (Zhou et al., 2025; San Roman et al., 2025; Wang et al., 2025). For diffusion-based generative models, watermarks can be embedded in the latent space and diffusion process to achieve audio generation with inherent watermarks (Liu et al., 2024b; Tang, 2025). Although these methods involve watermarking, they must run on specific generative models, whereas our focus is on general audio watermarking for arbitrary existing audio, making experimental comparisons infeasible due to differing application scenarios.

## 3 CONTENT ALIGNMENT DEGREE

To quantify how well the watermark aligns with audio content, we introduce the CAD metric. The intuition is to measure the proportion of the watermark that lies within energy-dense regions of the spectrogram. Unlike set-symmetric metrics such as intersection over union (Yu et al., 2016), CAD measures containment of watermark regions within content regions. Formally, CAD is defined as:

$$\text{CAD} = \frac{|W \cap C|}{|W|}, \tag{1}$$

where $W$ denotes the set of watermark regions, $C$ the set of audio content regions, and $|W \cap C|$ their intersection. The following steps outline our specific implementation.

**Step 1: Watermark Spectrogram Extraction**. The watermark waveform $x_w \in \mathbb{R}^T$ is obtained by subtracting the original audio $x \in \mathbb{R}^T$ from the watermarked audio $\hat{x} \in \mathbb{R}^T$:

$$x_w = \hat{x} - x. \tag{2}$$

Using STFT with 512 FFT points, hop length 128, and window length 512, we compute magnitude spectrograms $\mathbf{m}_w, \mathbf{m} \in \mathbb{R}^{f \times l}$ for $x_w$ and $x$, with $f = 257$ frequency bins and $l = T/128$ frames.

**Step 2: Normalization**. To align the energy scales of watermark and content spectrograms, we apply frame-wise min–max normalization:

$$\hat{\mathbf{m}}_w(i,j) = \frac{\mathbf{m}_w(i,j) - \min_i \mathbf{m}_w(i,j)}{\max_i \mathbf{m}_w(i,j) - \min_i \mathbf{m}_w(i,j)}, \quad \hat{\mathbf{m}}(i,j) = \frac{\mathbf{m}(i,j) - \min_i \mathbf{m}(i,j)}{\max_i \mathbf{m}(i,j) - \min_i \mathbf{m}(i,j)}. \tag{3}$$

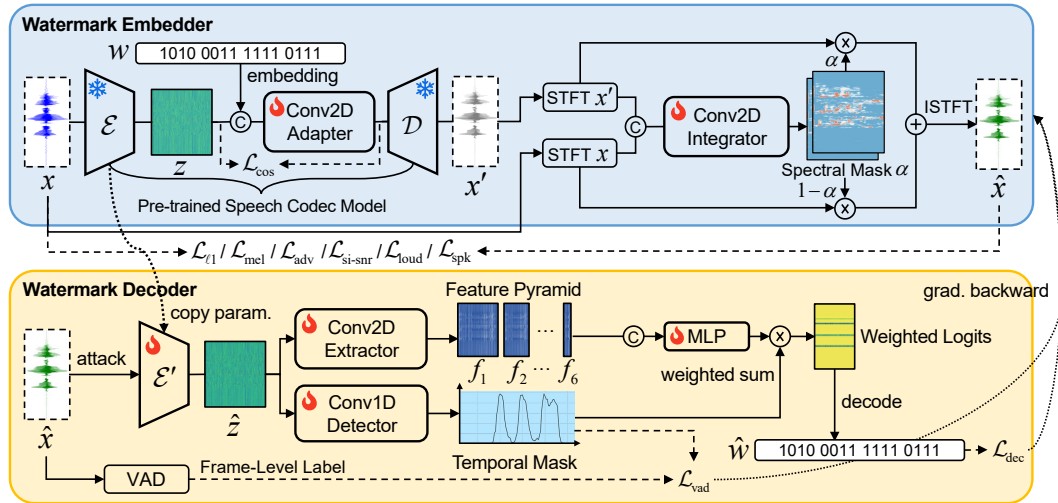

Figure 2: The overall architecture of our proposed AlignMark.

**Step 3: Binary Masking and Region Definition**. The frequency axis is divided into overlapping bands (window size 8, 50% overlap). For each band $k$, we average across frequencies to obtain band-specific $\hat{\mathbf{m}}_w^{(k)} \in \mathbb{R}^l$ and $\hat{\mathbf{m}}^{(k)} \in \mathbb{R}^l$. Binary masks are then formed using mean-thresholding:

$$\mathbf{b}_w^{(k)} = (\hat{\mathbf{m}}_w^{(k)} > \mu_w^{(k)}), \quad \mathbf{b}^{(k)} = (\hat{\mathbf{m}}^{(k)} > \mu^{(k)}), \tag{4}$$

where $\mu_w^{(k)}$ and $\mu^{(k)}$ are mean values of $\hat{\mathbf{m}}_w^{(k)}$ and $\hat{\mathbf{m}}^{(k)}$. The watermark and content sets are

$$W = \bigcup_{k=1}^{K} \{j : \mathbf{b}_w^{(k)}(j) = 1\}, \quad C = \bigcup_{k=1}^{K} \{j : \mathbf{b}^{(k)}(j) = 1\}. \tag{5}$$

**Step 4: CAD Computation**. In practice, directly computing $W \cap C$ from binary masks can be numerically sensitive in sparse spectrogram regions. To address this, we derive an equivalent form:

$$\text{CAD} = \frac{|W \cap C|}{|W|} = \frac{|W \cap (W \cap C)|}{|W \cup (W \cap C)|}, \tag{6}$$

where we define $I = W \cap C$. To approximate $I$ stably, we compute a continuous intersection map:

$$\mathbf{i} = \hat{\mathbf{m}}_w \odot \hat{\mathbf{m}}, \tag{7}$$

where $\odot$ is element-wise multiplication. The same band-averaging and thresholding procedure is then applied to obtain binary masks $\mathbf{b}_i^{(k)}$. CAD is finally computed using the equivalent form:

$$\text{CAD} = \frac{1}{K} \sum_{k=1}^{K} \frac{\sum (\mathbf{b}_w^{(k)} \odot \mathbf{b}_i^{(k)})}{\sum \max(\mathbf{b}_w^{(k)}, \mathbf{b}_i^{(k)})}. \tag{8}$$

Higher CAD values indicate that watermark regions are largely contained within content regions, whereas lower values suggest misalignment with energy-dense regions. CAD provides a quantitative measure of the degree to which the watermark is integrated into the content, offering valuable guidance for designing robust audio watermarking methods.

## 4 ALIGNMARK

The architecture of AlignMark, shown in Figure 2, includes a watermark embedder and a watermark decoder. Spectral and temporal masking are jointly trained to support content-aligned watermarking.

## 4.1 WATERMARK EMBEDDER

The watermark embedder comprises a frozen codec model (Zhang et al., 2024) with the quantization layer removed, an adapter $\mathcal{A}$, and an integrator $\mathcal{I}$. Following Li et al. (2025), the adapter embeds watermarks into speech latents, promoting their integration with the audio content. The integrator predicts spectral masks to integrate the intermediate audio from the codec with the original audio, generating the watermarked audio. This process is guided by perceptual losses and a VAD-based loss, encouraging the embedder to align the watermark with the audio content.

Specifically, given the original audio $x \in \mathbb{R}^T$, the codec encoder extracts speech latents $z \in \mathbb{R}^{d \times t}$, where $t$ is the number of frames and $d$ is the latent space dimension. The $n$-bit watermark $w \in \{0,1\}^n$ is transformed into $n$ embedding vectors via an embedding layer, summed to form $w_e \in \mathbb{R}^d$, which is broadcast across all frames and concatenated with $z$ to produce $z_w \in \mathbb{R}^{2d \times t}$. The adapter $\mathcal{A}$, a 6-layer 2D convolutional network, transforms $z_w$ into modified speech latents $z' \in \mathbb{R}^{d \times t}$, which are decoded by the codec decoder to generate the intermediate audio $x' \in \mathbb{R}^T$.

Both $x'$ and $x$ are converted into spectrograms $\mathbf{s}_{x'}$ and $\mathbf{s}_x \in \mathbb{R}^{2f \times l}$ via STFT, where $f$ is the number of frequency bins, $l$ is the number of frames, and $2f$ is the concatenation of the complex real and imaginary components. The spectrograms are concatenated to form $\mathbf{s}_c \in \mathbb{R}^{4f \times l}$, which is processed by a 4-layer 2D convolutional embedder to predict a spectral mask $\alpha \in \mathbb{R}^{2f \times l}$. The watermarked spectrogram $\mathbf{s}_w \in \mathbb{R}^{2f \times l}$ is obtained by combining $\mathbf{s}_{x'}$ and $\mathbf{s}_x$ using $\alpha$:

$$\mathbf{s}_w = \mathbf{s}_{x'} \cdot \alpha + \mathbf{s}_x \cdot (1 - \alpha). \tag{9}$$

Finally, the inverse STFT is applied to $\mathbf{s}_w$ to produce the watermarked audio $\hat{x} \in \mathbb{R}^T$. During training, the VAD-based loss constrains the spectral mask to embed the watermark in voiced frames, while leaving silent frames almost unchanged. For the frequency dimension, $\hat{x}$ is compared to $x$ using perceptual losses, including cosine similarity between $z$ and $z'$, speaker similarity[1], and psychoacoustic-based loudness loss San Roman et al. (2024). These losses penalize the watermark in non-content frequency bands from semantic, acoustic, and perceptual perspectives, encouraging the spectral mask to align with the audio content.

## 4.2 WATERMARK DECODER

The watermark decoder consists of a detector for predicting the temporal mask and an extractor for capturing feature pyramids. The temporal mask is used to filter voiced frames from the feature pyramids, which are then utilized to decode the watermark. During training, $\hat{x}$ undergoes various differentiable attacks to produce $\tilde{x}$, including standard audio distortions (San Roman et al., 2024; Li et al., 2025) (replace, mask, shuffle, compression, filter, pitch shift) and neural transformations (codec (Défossez et al., 2022), vocoder (Siuzdak, 2024)). These attacks serve as data augmentation for model optimization. The detailed decoding process follows these steps:

**Feature Pyramid Extraction**. The attacked audio $\tilde{x}$ is fed into a feature encoder, initialized with the codec encoder's parameters, to extract speech latents $\hat{z} \in \mathbb{R}^{d \times t}$. These latents are processed by a 6-layer 2D convolutional extractor, where each layer downsamples the channel dimension and refines intermediate features. At scale $i$, the feature $\hat{z}_i \in \mathbb{R}^{c_i \times d/2^i \times t}$ is computed iteratively as $\hat{z}_i = \text{Conv2D}_i(\hat{z}_{i-1})$, with $\hat{z}_0 = \hat{z}$ as input, $c_i = 2^{i-1} \cdot 16$ as the channel dimension, and $\text{Conv2D}_i(\cdot)$ representing the $i$-th convolutional layer. Each $\hat{z}_i$ is processed by a fully connected layer to produce the feature pyramid $f_i \in \mathbb{R}^{c_i \times t}$, reducing the $d/2^i$ dimension. The final representation $f \in \mathbb{R}^{c \times t}$ is obtained by concatenating $f_i$ across all scales, with $c = \sum_{i=1}^{6} c_i$.

**Temporal Mask Prediction.** A 4-layer 1D convolutional detector predicts a frame-wise temporal mask $p \in \mathbb{R}^t$, where values range from 0 to 1, with higher values indicating a greater probability of voiced frames. The prediction process can be expressed as $p = \text{Sigmoid}(\text{Conv1D}(\hat{z}))$, where $\text{Conv1D}(\cdot)$ denotes the 1D convolution detector, and $\text{Sigmoid}(\cdot)$ maps the output to $[0, 1]$.

**Watermark Decoding**. The feature pyramid $f$ is converted into frame-wise logits $w_f \in \mathbb{R}^{(n/4) \times 16 \times t}$ using a 2-layer MLP. The $(n/4) \times 16$ format converts an $n$-bit binary watermark into hexadecimal representation, following (Li et al., 2025), to stabilize training. Finally, the temporal mask $p$ directly filters these logits via a weighted sum, $\hat{w} = \sum_t (w_f \cdot p)$. The weighted logits

---
[1]https://github.com/resemble-ai/Resemblyzer

$\hat{w} \in \mathbb{R}^{(n/4) \times 16}$ are processed with an argmax operation and converted back to binary format, yielding the final $n$-bit decoded watermark. The temporal mask encourages the extractor to focus on audio content for watermark extraction and also guides the embedder to align with voiced frames.

### 4.3 TRAINING LOSS

We incorporate multiple perceptual losses to preserve audio quality. Standard losses (San Roman et al., 2024) include L1 ($\mathcal{L}_{\ell 1}$), Mel spectrogram ($\mathcal{L}_{\text{mel}}$), adversarial ($\mathcal{L}_{\text{adv}}$), SI-SNR ($\mathcal{L}_{\text{si-snr}}$), and time-frequency loudness ($\mathcal{L}_{\text{loud}}$). In addition, we introduce the following losses: the speaker similarity loss $\mathcal{L}_{\text{spk}}$, which preserves speaker characteristics by minimizing the distance between Resemblyzer embeddings; the latent cosine loss $\mathcal{L}_{\text{cos}}$, which constrains modifications in the latent space to ensure minimal distortion; the VAD-based loss $\mathcal{L}_{\text{VAD}}$, which supervises temporal masks using binary cross-entropy; and the decoding loss $\mathcal{L}_{\text{dec}}$, which applies cross-entropy to hexadecimal classification.

Formally, these losses can be written as:

$$\mathcal{L}_{\text{spk}} = 1 - \cos(\text{Emb}(x), \text{Emb}(\hat{x})), \qquad \mathcal{L}_{\text{cos}} = 1 - \cos(z, z'),$$

$$\mathcal{L}_{\text{vad}} = -\frac{1}{t}\sum_{i=1}^{t}\big[v_i \log p_i + (1 - v_i)\log(1 - p_i)\big], \quad \mathcal{L}_{\text{dec}} = -\frac{1}{n/4}\sum_{j=1}^{n/4}\sum_{k=1}^{16} y_{jk}\log \hat{w}_{jk}, \tag{10}$$

where $x$ and $\hat{x}$ are the original and watermarked audio, $\text{Emb}(\cdot)$ extracts speaker embeddings, $z$ and $z'$ are the latents before and after adaptation, $v_i \in \{0, 1\}$ is the VAD label following Li et al. (2025) (0 for silent/masked/replaced frames, 1 otherwise), $p_i$ is the predicted temporal mask, and $y_{jk}$ and $\hat{w}_{jk}$ are the one-hot label and predicted probability for the $j$-th hexadecimal digit.

The total loss is a weighted sum of all terms:

$$\mathcal{L}_{\text{total}} = \sum_{\ell \in \mathcal{L}} \lambda_\ell \mathcal{L}_\ell \tag{11}$$

where $\mathcal{L}$ denotes all loss terms with weights: $\lambda_{\ell 1} = 0.01$, $\lambda_{\text{mel}} = 0.1$, $\lambda_{\text{adv}} = 0.5$, $\lambda_{\text{si-snr}} = 0.01$, $\lambda_{\text{loud}} = 0.1$, $\lambda_{\text{spk}} = 0.1$, $\lambda_{\text{cos}} = 0.1$, $\lambda_{\text{vad}} = 1.0$, $\lambda_{\text{dec}} = 4.0$. This configuration is designed to balance the scales of the losses and enhance the decoding loss to accelerate convergence.

## 5 EXPERIMENTS

### 5.1 EXPERIMENTAL SETUPS

**Training**. The codec model uses weights from SpeechTokenizer[2]. The adapter $\mathcal{A}$ employs skip-gated blocks (Liu et al., 2024a) with layers of 32 channels. The integrator $\mathcal{I}$ consists of STFT (256 FFT points, hop length 64, window length 256) and 2D convolutions (64 channels, LeakyReLU with slope 0.1 (Maas et al., 2013)). The watermark detector and extractor employ different architectures: the detector uses 1D convolutions (256 channels, GELU), while the extractor uses 2D convolutions (kernel $(5, 3)$, stride $(2, 1)$, padding $(0, 1)$, channels doubling from 16, GELU (Hendrycks & Gimpel, 2016)). The watermark bit length $n$ is 16. Adam optimizer (Kingma, 2014) is used with a $5e^{-5}$ learning rate, trained for 300 epochs, selecting the checkpoint with the lowest loss.

**Dataset**. Our experiments use three datasets: VCTK (Yamagishi, 2012), LibriSpeech (Panayotov et al., 2015), and LJSpeech (Ito & Johnson, 2017). For VCTK, 200 audio samples are randomly selected for testing, with the remaining samples used as the training set. Similarly, 200 audio samples are randomly selected from LibriSpeech and LJSpeech respectively, forming a 600-sample test set.

**Metrics and Baselines**. We evaluate robustness using our proposed CAD, ACC, and false attribution rate (FAR). ACC measures the ratio of correctly decoded bits. FAR is computed by comparing each decoded watermark to one positive and 599 negative test watermarks via Hamming distance, representing the proportion of cases where the closest match is not the positive sample. Audio quality is assessed through objective metrics (PESQ (Rix et al., 2001), SI-SNR, STOI (Taal et al., 2010), NISQA (Mittag et al., 2021)) and subjective ABX tests. NISQA scores (1-5) are obtained via

---

[2]https://huggingface.co/fnlp/SpeechTokenizer

Table 1: Evaluation of robustness against complex attacks (ACC ↑, FAR ↓, **bold** for best)

| Attack | WavMark | | AudioSeal | | Timbre | | VoiceMark | | AlignMark | |
|---|---|---|---|---|---|---|---|---|---|---|
| | ACC | FAR | ACC | FAR | ACC | FAR | ACC | FAR | ACC | FAR |
| Traditional Distortion | | | | | | | | | | |
| Resample | 1.00 | 0.00 | 1.00 | 0.00 | 1.00 | 0.00 | 0.99 | 0.06 | **1.00** | **0.00** |
| Boost Volume | 1.00 | 0.00 | 1.00 | 0.00 | 1.00 | 0.00 | 0.98 | 0.09 | **1.00** | **0.00** |
| Duck Volume | 1.00 | 0.00 | 1.00 | 0.00 | 1.00 | 0.00 | 0.98 | 0.07 | **1.00** | 0.01 |
| Highpass Filter | 1.00 | 0.00 | 1.00 | 0.00 | 1.00 | 0.00 | 0.99 | 0.05 | **1.00** | **0.00** |
| Lowpass Filter | 1.00 | 0.00 | 1.00 | 0.00 | 1.00 | 0.02 | 0.76 | 0.82 | **1.00** | **0.00** |
| Bandpass Filter | 1.00 | 0.01 | 1.00 | 0.00 | 0.99 | 0.06 | 0.76 | 0.72 | **1.00** | **0.00** |
| AAC Compression | 1.00 | 0.00 | 0.73 | 0.96 | 1.00 | 0.00 | 0.98 | 0.07 | 0.99 | 0.02 |
| MP3 Compression | 0.98 | 0.04 | 1.00 | 0.00 | 1.00 | 0.00 | 0.85 | 0.56 | 0.99 | 0.01 |
| Echo | 0.99 | 0.02 | 1.00 | 0.00 | 1.00 | 0.00 | 0.98 | 0.11 | **1.00** | 0.01 |
| Crop | 0.98 | 0.03 | 0.62 | 0.91 | 1.00 | 0.00 | 0.98 | 0.08 | 0.99 | 0.02 |
| Pink Noise | 0.98 | 0.05 | 1.00 | 0.01 | 1.00 | 0.01 | 0.99 | 0.05 | **1.00** | **0.00** |
| Gassuion Noise | 0.51 | 1.00 | 0.77 | 0.73 | 0.87 | 0.60 | 0.54 | 0.99 | **0.99** | **0.02** |
| Smooth | 0.98 | 0.03 | 1.00 | 0.00 | 1.00 | 0.00 | 0.76 | 0.72 | 0.99 | 0.03 |
| Pitch Shifting | 0.52 | 0.97 | 0.53 | 0.97 | 0.54 | 0.72 | 0.85 | 0.50 | **1.00** | **0.00** |
| Speed Change | 0.51 | 0.99 | 0.50 | 0.99 | 0.50 | 0.98 | 0.53 | 0.95 | **0.94** | **0.21** |
| Neural Transformation | | | | | | | | | | |
| EnCodec | 0.51 | 1.00 | 0.50 | 1.00 | 0.57 | 0.99 | 0.96 | 0.16 | **0.99** | **0.02** |
| FACodec | 0.51 | 1.00 | 0.50 | 0.99 | 0.53 | 1.00 | 0.88 | 0.45 | **0.93** | **0.24** |
| SpeechTokenizer | 0.51 | 1.00 | 0.50 | 0.99 | 0.57 | 0.99 | 0.95 | 0.21 | **0.96** | **0.11** |
| Vocos | 0.51 | 1.00 | 0.50 | 1.00 | 1.00 | 0.00 | 0.99 | 0.04 | **1.00** | **0.00** |
| HiFiGAN | 0.51 | 1.00 | 0.50 | 1.00 | 0.92 | 0.48 | 0.94 | 0.22 | **0.99** | **0.01** |
| Denoise | 0.51 | 1.00 | 0.67 | 0.79 | 0.78 | 0.65 | 0.76 | 0.60 | **0.92** | **0.27** |
| Average | 0.78 | 0.45 | 0.79 | 0.47 | 0.86 | 0.32 | 0.87 | 0.37 | **0.98** | **0.05** |
| CAD ↑ | 0.54 | | 0.58 | | 0.67 | | 0.72 | | **0.85** | |

automated evaluation, with higher scores indicating better naturalness. For ABX tests, 20 subjects perform 10 trials per method, identifying whether a randomly selected sample X matches the original A or watermarked B. Scores near 0.5 indicate imperceptible watermarks.

We compare against four state-of-the-art methods with publicly available implementations: Wav-Mark (Chen et al., 2023), AudioSeal (San Roman et al., 2024), Timbre (Liu et al., 2024a), and VoiceMark (Li et al., 2025). All methods use 16-bit watermarks except Timbre (10 bits). For WavMark, undetected watermarks default to zeros. These baselines employ diverse architectures, providing a comprehensive evaluation of our approach.

**Attack Scenarios**. We evaluate the robustness of our method across 21 attack scenarios, including 15 traditional audio distortions (San Roman et al., 2024; O'Reilly et al., 2025) and 6 neural transformations (Zhao et al., 2025; Défossez et al., 2022; Zhang et al., 2024; Ju et al., 2024; Kong et al., 2020; Siuzdak, 2024). Among these, AAC/MP3 compression, denoising (Zhao et al., 2025), FA-Codec (Ju et al., 2024), SpeechTokenizer (Zhang et al., 2024), and HiFiGAN (Kong et al., 2020) are unseen during training; detailed parameters are provided in Appendix A. Attacks that entirely alter content, such as using watermarked audio as a prompt to generate new audio via text-to-speech, are excluded from our experiments, as they fall outside the scope of protecting the audio content's IP.

## 5.2 ROBUSTNESS EVALUATION

**Robustness to Traditional Distortion**. As shown in Table 1, AlignMark outperforms other methods in ACC and FAR across most traditional distortions, especially in pitch shifting, speed changes, and Gaussian noise attacks. Pitch shifting and speed change results demonstrate the effectiveness of the feature pyramid, while noise resistance validates the robustness of content-aligned watermarking. AlignMark shows slightly lower performance against compression and cropping attacks, as these directly damage portions of audio content, consequently disrupting the embedded watermarks. This issue could be mitigated by introducing more augmentations or using larger datasets.

**Robustness to Neural Transformation**. Across all neural transformation attacks in Table 1, Align-Mark significantly outperforms the baselines in both ACC and FAR, especially under denoising, where the ACC of all baselines drops below 0.8, while AlignMark maintains above 0.92. The results demonstrate that both denoising and reconstruction attacks preserve audio content to some extent, providing content-aligned watermarking with inherent robustness. Overall, as shown in Figure 1, the CAD metric shows a positive correlation with the average ACC and performance under neural transformations, further validating the connection between CAD and robustness.

**Robustness to Denoising Levels**. Following previous work (O'Reilly et al., 2025), we first apply Gaussian noise at SNRs of 20dB, 15dB, 10dB, 5dB, and 0dB, respectively, and then perform denoising (Zhao et al., 2025) to remove the watermark at different levels. The results in Figure 3 show that all baselines experience a significant ACC drop as SNR decreases, with WavMark completely failing to decode the watermark. In contrast, AlignMark demonstrates remarkable resilience, showing no noticeable drop in ACC from 20dB to 15dB SNR, and maintaining an ACC of 0.8 even at 0dB SNR, outperforming all baselines. The results demonstrate that aligning watermarks with audio content enhances their resilience against separation and removal.

**Robustness to Pitch Shifting Levels**. Figure 4 shows ACC of different methods under varying levels of pitch shifting. From semitones -1 to 1, AlignMark consistently achieves robust performance with an ACC of 1.0. In contrast, most other methods fail to decode, except for VoiceMark, which maintains an ACC above 0.6. Timbre exhibits a strong dependence on fixed frequency bands, leading to watermark reversal (near-zero ACC) at semitone shifts of -1 and 1. AlignMark introduces a feature pyramid to enhance decoding robustness and thereby overcomes this limitation.

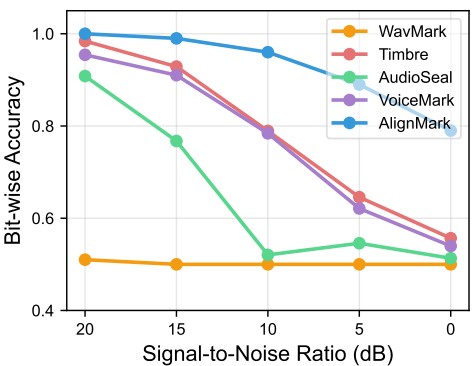
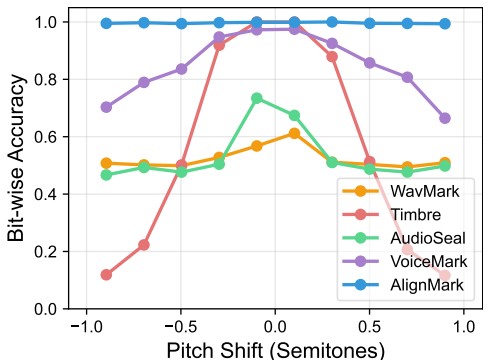

Figure 3: Robustness to denoising levels.    Figure 4: Robustness to pitch shifting levels.

## 5.3 AUDIO QUALITY

Table 2 summarizes audio quality results. We group methods into signal-based and synthesis-based models. Signal-based models embed watermarks directly into waveforms or spectrograms, yielding strong objective metrics but lower NISQA naturalness. Synthesis-based models, including Voice-Mark and our AlignMark, leverage codec models for watermark embedding, with three codec-only models as baselines. Compared to codec baselines such as EnCodec and SpeechTokenizer, Align-Mark achieves superior objective performance and higher NISQA naturalness than most watermarking methods. Its ABX score of 0.51, close to 0.5, further indicates imperceptible quality degradation for human listeners.

## 5.4 ABLATION STUDY

We conduct ablation on each key component of AlignMark, with average results shown in Table 3, where removing any component degrades either robustness or audio quality. Removing the spectral mask reduces both robustness and audio quality. The temporal mask also contributes to both robustness and audio quality. Removing the feature pyramid causes training collapse, with over-optimization of audio quality and reduced watermark capacity. Although CAD remains high, the decoder fails to converge, leading to decoding failure. This suggests that CAD is meaningful only with a well-trained decoder, indicating the importance of a appropriate decoder design.

Table 2: Audio quality evaluation. ABX is better when closer to 0.5 (95% confidence interval).

| Method | PESQ ↑ | STOI ↑ | SI-SNR ↑ | NISQA ↑ | ABX |
|---|---|---|---|---|---|
| *Signal-based Model* | | | | | |
| WavMark | 4.10 | 0.99 | 36.87 | 4.21 | 0.50±0.08 |
| Timbre | 3.72 | 0.99 | 23.93 | 4.22 | 0.57±0.11 |
| AudioSeal | 4.37 | 0.99 | 27.60 | 4.28 | 0.49±0.07 |
| *Synthesis-based Model* | | | | | |
| EnCodec | 2.82 | 0.92 | 5.67 | 3.97 | - |
| FACodec | 2.93 | 0.94 | 3.91 | 4.41 | - |
| SpeechTokenizer | 2.67 | 0.92 | 1.79 | 4.28 | - |
| VoiceMark | 2.19 | 0.90 | 1.90 | 4.36 | 0.72±0.09 |
| AlignMark | 3.03 | 0.95 | 12.16 | 4.31 | 0.51±0.10 |

Table 3: Ablation study.

| Method | ACC ↑ | FAR ↓ | PESQ ↑ | STOI ↑ | CAD ↑ |
|---|---|---|---|---|---|
| AlignMark | 0.98 | 0.05 | 3.03 | 0.95 | 0.85 |
| w/o Temporal mask | 0.80 | 0.49 | 2.75 | 0.94 | 0.70 |
| w/o Spectral mask | 0.84 | 0.40 | 1.89 | 0.85 | 0.76 |
| w/o Feature pyramid | 0.51 | 0.97 | 4.64 | 1.00 | 0.86 |

## 5.5 WATERMARK-CONTENT ALIGNMENT ANALYSIS

On different audio samples, we visualize the watermark spectrograms averaged over time and reduced using PCA to analyze the alignment between the watermark and audio content. As shown in Figure 5, AlignMark's watermarks vary across different audio, showing an association with the audio content. In contrast, other methods produce clustered watermarks with limited association. This result provides additional evidence of AlignMark's ability to achieve content-aligned watermarking.

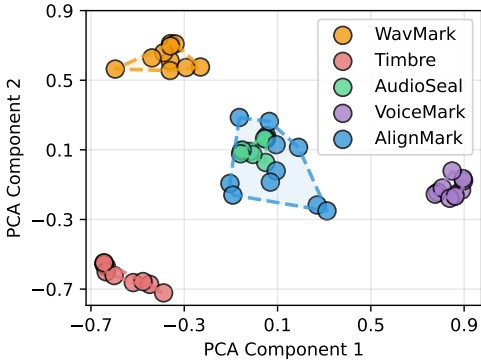

Figure 5: PCA-based visualization of time-averaged watermark spectrograms.

## 6 CONCLUSION

In this work, we investigate the relationship between watermark robustness and audio content alignment, introducing the content alignment degree (CAD) metric and empirically observing a positive correlation with robustness. Guided by CAD, we propose AlignMark, a content-aligned audio watermarking method. AlignMark leverages spectral masking in the embedder, and temporal masking along with a feature pyramid in the decoder, combined with multiple perceptual losses, to explicitly align watermark embedding with audio content, enhancing robustness against diverse distortions and transformations while preserving perceptual quality. Extensive experiments on three datasets and 21 attack scenarios show that AlignMark achieves state-of-the-art performance, with average ACC 0.98 and FAR 0.05, while maintaining imperceptible impact on audio quality for human listeners.

ETHICS STATEMENT

To evaluate the subjective perception of AlignMark's impact on audio quality, we conducted human subjective testing experiments, including ABX tests, as detailed in the experimental section. All recruited participants provided informed consent, and their responses were used solely for academic research purposes. No personal information beyond the questionnaire content was collected, and strict confidentiality was maintained regarding their answers. Additionally, all audio samples used in the subjective tests were uniformly discarded after the experiments to prevent any risks associated with data leakage. While AlignMark aims to protect intellectual property, we acknowledge the potential for misuse, such as embedding unauthorized watermarks or circumventing watermark detection. This work is intended strictly for lawful and academic applications, and we encourage future research to explore safeguards against unethical use.

REPRODUCIBILITY STATEMENT

To ensure the reproducibility of our results, we provide a detailed description of the experimental setup, including datasets, model architectures, hyperparameters, and evaluation metrics. The pre-trained models, training configurations, and attack scenarios are clearly documented in the main text and appendix. Additionally, we commit to releasing the code, pre-trained weights, and data preprocessing scripts upon publication to facilitate replication and further research. We encourage the community to validate and extend our findings under diverse experimental conditions.

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

## A  ATTACK PARAMETERS

Here are the settings of attack scenarios used in our experiments. The random seed is fixed across all experiments:

- **Resample**: Upsample from 16kHz to 32kHz and then downsample back to 16kHz.
- **Boost Volume**: Increase volume by 20%.
- **Duck Volume**: Decrease volume by 20%.
- **Highpass Filter**: Apply a highpass filter to remove frequencies below 500Hz.
- **Lowpass Filter**: Apply a lowpass filter to remove frequencies above 4000Hz.
- **Bandpass Filter**: Allow frequencies between 500Hz and 4000Hz to pass through.
- **AAC Compression**: Apply AAC compression at 64kbps.
- **MP3 Compression**: Apply MP3 compression at 32kbps.
- **Echo**: Add an echo effect with random delay (0.1-0.5s) and random volume (0.1-0.5).
- **Crop**: Randomly retain 80% of the audio by cropping out the remaining 20%.
- **Pink Noise**: Add pink noise with fixed standard deviation of 0.1.
- **Gaussian Noise**: Add Gaussian noise with SNR set to 10dB.
- **Smooth**: Apply a moving average filter with a random window size between 2 and 10.
- **Pitch Shifting**: Randomly shift pitch within semitones [-1, 1].
- **Speed Change**: Randomly change speed by resampling with a factor between 0.5 and 2.0.
- **EnCodec**: Reconstruct audio using the pre-trained model from Défossez et al. (2022).
- **FACodec**: Reconstruct audio using the pre-trained model from Ju et al. (2024).
- **SpeechTokenizer**: Reconstruct audio using the pre-trained model from Zhang et al. (2024).

- **Vocos**: Reconstruct audio using the pre-trained model from Siuzdak (2024).
- **HiFiGAN**: Reconstruct audio using the pre-trained model from Kong et al. (2020).
- **Denoise**: Add Gaussian noise at SNRs of 20dB, 15dB, 10dB, 5dB, and 0dB, then apply denoising (Zhao et al., 2025) to remove the watermark at different levels. Finally, average the evaluation metrics across all levels.

## B  VISUALIZATIONS

Figure 6 visualizes the spectrograms of the intermediate audio generated by the codec model and the spectral mask to provide a detailed analysis. In Figure 6 (a), the intermediate audio exhibits a spectrogram that resembles natural speech, allowing it to be seamlessly embedded into the audio. This encourages the content-aligned watermark embedder to embed watermarks within speech regions rather than introducing artifacts into the background. Figures 6 (b–c) demonstrate that the spectral mask aligns with audio content while avoiding the fundamental frequency regions that could severely degrade audio quality, thereby preserving perceptual quality. These results confirm that AlignMark achieves content-aligned watermarking without introducing background artifacts as seen in previous methods, while minimizing the impact on audio quality.

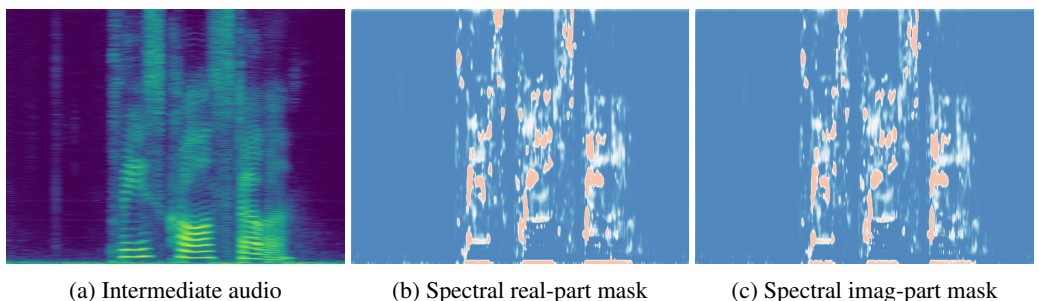

| (a) Intermediate audio | (b) Spectral real-part mask | (c) Spectral imag-part mask |

Figure 6: Visualization of intermediate audio and spectral mask. The spectral mask (red regions indicate embedding areas) accurately aligns with audio content while avoiding the fundamental frequency regions that could severely degrade audio quality.

## C  THE USE OF LARGE LANGUAGE MODELS

Large language models were used to refine this paper's writing for accurate spelling and grammar.

