# OpenReview forum: "AlignMark: Content-Aligned Audio Watermarking for Robustness Against Neural Transformations"
_ICLR.cc/2026/Conference — ICLR 2026 Conference Withdrawn Submission_

### Official Review · Reviewer_fSoP · 2025-10-28

**Soundness:** 1
**Presentation:** 2
**Contribution:** 2
**Rating:** 2
**Confidence:** 4

**Summary:**

This paper proposes two items. The first is content alignment degree(CAD), a metric measures how well the watermark spectrogram aligns with content spectrogram. The second is AlignMark, a watermarking scheme that considers content alignment when embedding watermark.

However, I am not convinced by the evaluation result due to several concerns (listed in "Weakness") in the evaluation setup. This means the robustness of proposed method is not verified yet and remain undetermined.

**Strengths:**

The feature pyramid and the use of VAD are sounding ideas. The presentation of the overall workflow is clear and comprehensive.

**Weaknesses:**

Major concerns:

Most of the major concerns are about evaluation. Proposed method has seen the attacks during training including EnCodec and Vocoder (ref. L249-L251). It's unclear if these attacks have different parameters at inference time. It has been shown by past works (WavMark, AudioSeal, etc.) that seen attacks can usually be well addressed at inference time, but unseen attacks are not. For example, AudioSeal used a different set of attacks at inference time.

The evaluation is not conducted in a well-controlled configuration. There's no valid control about quality of the watermarked signal, neither the control to make sure the bit capacity is the same across different methods. Such control can be found in for example WavMark (bit-per-second) and AudioSeal (ViSQOL, MUSHRA). The robustness comparison is fair only if at least one control is implemented, such as all the methods are either tuned to have the same perceptual quality, the same amount of information to embed, or the same amount of freedom to change the target signal.

The ABX test could be argued to serve the role. However, it's result is somehow contradicts with other metrics. For example, Timbre has 3.72 on PESQ and 23.93 on SI-SNR and ABX result of 0.57, indicating some degree of perceptuality. However, proposed method scored 3.03 on PESQ and 12.16 on SI-SNR, but get an ABX result of 0.51, which is unusual and deserve further investigation.

In addition, based on the description on L226-L227, it seems w_e is broadcasted across all frames. It means, the bit-per-second of proposed method can be lower than WavMark (5.33bps ver.) when the audio is more than 3 seconds. This makes the robustness comparison lose its ground because baseline methods are not configured in the same condition. Moreover, it's somehow unconvincing that 12.16db SI-SNR is "imperceptible" on speech. Most of watermarks (including those cited in this work) can reach higher than 30db and still sometimes being complained that they are perceptible by industrial audio engineers.

Furthermore, the strengths of robustness attack are not properly determined, for example, a speech change of 0.5/2.0 is a really extreme case. If the content is under extremely heavy distortion, it's no longer meaningful to discuss the "robustness of watermark" due to the content it has to protect is already destroyed. Again, an example from previous work: Wen et al. selected the strength of attack based on VISQOL score (>4 or >3) to ensure the strengths of attacks fall within a reasonable range.

Minor concerns:
- This work discussed about intellectual property protection. However, based on the VAD in AlignMark and the design of CAD, seems to be focusing on clean speech signal. It's unclear how could this methodology being applied in generic audio content that clear activation is not always there.

- The main argument of this work is, embedded watermark must be coupled with the content, need to be clarified more. Otherwise, all the existing works employing psychoacoustic models could be argued already satisfying such claim because these models tend to put information in the sidelobe of a main spectral component, under its masking curve, thus always "coupled" with the content.

- L066: It seems Fig. 1 is a result of robustness evaluation, perhaps this refers to the main result in Table 1, but it's not mentioned explicitly.

- L083-084: If a watermark considers perceptual loss, then it has already distinguished voiced and silent contents, because making any change in silent segments breaks the perceptual masking curve.

- L098-099: If I understand correctly, Wen et al. only mentioned that most of methods are dependent on "frequency-based patterns", not fixed bands.

- L198-199: "W \cap C" is not the same intersection as the rest. Because "i" computed on "m", where the rest of operations computed on "b". They should not be written as the same operation in Eq.6.

- Suggest using AUC than FAR if only one can be chosen, as it's not limited to the test set size. Also, the 600-class classification with hamming distance is hard to be scaled to real applications.

- L324: In Table 1, if multiple methods reached best performances, all of them should be marked in bold. Also, when proposed method is not the best, the best baseline should be marked in bold. Otherwise, it's somehow misleading.

- L344: In Table 1, AudioSeal's performance against EnCodec attack is pretty low. However, EnCodec is part of AudioSeal plus it has seen EnCodec during its training and shown robustness according to EnCodec according to original paper. Another evaluation paper from another group of researchers also observed that AudioSeal has robustness against Encodec(https://arxiv.org/abs/2505.19663). I believe the result in Table 1 needs further investigation.

- L359: The message amount of Timbre is lower than other baselines, which is unfair. Anyway, this is also an example of not well-controlled scenario as mentioned above.

- L374-375: The ablation study in Table 3 reports only overall improvement. There's no direct evidence that feature pyramid benefits "speed change" and "pitch shifting" attacks.

- L392: What is the meaning of "separation" here? The text only mentioned denoising.

- L458: In Figure 5, it simply shows that AlignMark is located in the same place as AudioSeal with larger variation. What's the direct evidence of "vary across different audio, showing an association with the audio content" for AlignMark?


Overall, although I feel the proposed method could have some potential, but the current evaluation hardly provides clear evidence about its strength in robustness as claimed in the beginning. Further refinement of evaluation would be a nice step to proceed.

**Questions:**

- Does "f" on L230 the same as "f" on L157? I suppose they are not?

- Is Emb(.) a pre-trained embedder?

- What's the audio length in the train and test set? Are they pre-processed into equal sized chunks?

- What's the pitch shifting algorithm used for Figure 4? AudioSeal has shown that it is robust against 10% speed change, which means more than 1 semitone in pitch change.

---

### Official Review · Reviewer_AAXa · 2025-10-30

**Soundness:** 3
**Presentation:** 3
**Contribution:** 3
**Rating:** 6
**Confidence:** 3

**Summary:**

The authors propose to add watermarking to the region that contains semantic content of the audio to make it more robust under different kinds of distortions.

**Strengths:**

- The observation the authors made that adding watermark to the same region as semantic content is simple yet powerful. The experimental results show that the performance is really good when facing various distortions.

- The paper is really well-written and easy to follow.

**Weaknesses:**

- Most of the tested distortions are not designed with the prior that the watermarking might be added to the same region as content. Although I did not find any such adversarial attack in watermarking, there are some attack examples for synthetic speech [1] which can be adapted and used to test the robustness of the proposed watermarking scheme under malicious attacks.

- The proposed metric CAD does not distinguish a warkmark evenly smeared on whole content region or a sharp point inside the content region, although intuitively the robustness of these two patterns should be totally different.

[1] Can DeepFake Speech be Reliably Detected? Hongbin Liu, Youzheng Chen, Arun Narayanan, Athula Balachandran, Pedro J. Moreno, Lun Wang

**Questions:**

How will the proposed watermarking perform under attack designed targeting it?

---

### Official Review · Reviewer_djf5 · 2025-10-31

**Soundness:** 3
**Presentation:** 3
**Contribution:** 2
**Rating:** 2
**Confidence:** 4

**Summary:**

This paper introduces a Content Alignment Degree (CAD) metric to quantify watermark-content alignment and observes a positive correlation between CAD and robustness. Guided by this observation, the authors propose AlignMark, which employs spectral masking in the embedder, temporal masking in the decoder, multiple perceptual losses, and a feature pyramid architecture.

**Strengths:**

The CAD metric provides a fresh perspective by quantifying watermark quality through content alignment rather than traditional perceptual metrics. The paper identifies a limitation of existing methods: embedding watermarks as background artifacts makes them vulnerable to denoising attacks. This insight offers valuable understanding of robustness mechanisms in audio watermarking.

**Weaknesses:**

1.The CAD metric alone proves insufficient for reliably predicting model robustness, as evidenced by the ablation experiments in Table 3: removing the feature pyramid maintains CAD at 0.86 (comparable to 0.85 in the full model), yet triggers performance degradation with ACC collapsing from 0.98 to 0.51 and FAR escalating from 0.05 to 0.97. This paradox underscores the conditional validity of CAD, high alignment does not guarantee robust performance. CAD provides meaningful guidance only when paired with a well-trained decoder; improper decoder architecture (such as the absence of the feature pyramid) results in decoder convergence failure and complete decoding breakdown, despite maintaining high alignment scores.

2.This paper does not explicitly articulate how each component of AlignMark aligns with the CAD, leaving CAD's guiding role unclear. Specifically, Section 4 introduces spectral masks without clarifying their design goal of improving frequency-domain CAD; Section 4.3 presents loss functions without connecting how they constrain watermark distribution to enhance CAD.

3.This paper lacks concrete analysis for robustness against neural transformation.

4.Additionally, hyperparameters (window size, overlap, frequency bands) lack principled justification, appearing purely empirical. These issues undermine CAD's credibility as a robustness indicator.

5.PCA is introduced without a formal definition (or citation), and no parameter settings are reported for the PCA experiments in section 5.5.

6.The paper does not report computational costs, including training time, and inference latency, hindering practical deployment assessment.

7.Furthermore, watermark capacity is fixed at 16 bits with no evaluation of longer watermarks, limiting understanding of scalability and performance trade-offs.

**Questions:**

1.Why ALIGNMARK resists neural transformation? The paper demonstrates empirical robustness but fails to explain the underlying mechanism.

2.Does CAD appear to capture only content region overlap while neglecting the effectiveness of watermark distribution across multi-scale frequencies? The missing feature pyramid, for example, impairs cross-scale watermark extraction, yet this critical deficiency goes undetected by CAD. Visualizations of watermark distribution across frequency scales are needed to reveal limitations of CAD.

3.Determination logic of window length “8” and overlap rate “50%”: Are they derived from the frequency-resolution characteristics of the audio signal, or are they simply optimal values obtained through experimental?

---

### Note · Authors · 2025-12-30

I have read and agree with the venue's withdrawal policy on behalf of myself and my co-authors.